

# Psychometric properties of the Perceived Stress Scale (PSS): measurement invariance between athletes and non-athletes and construct validity

Yi-Hsiang Chiu[1,*], Frank Jing-Horng Lu[2,*], Ju-Han Lin[3], Chiao-Lin Nien[4], Ya-Wen Hsu[5] and Hong-Yu Liu[6]

[1] Graduate Institute of Physical Education, National Taiwan Sport University, Taoyuan, Taiwan
[2] Graduate Institute of Sport Coaching Science, Chinese Culture University, Taipei, Taiwan
[3] Department of Physical Education and Kinesiology, National Dong Hwa University, Hua Lien, Taiwan
[4] Department of Physical Education, National Taiwan University of Sport, Taichung, Taiwan
[5] Department of Physical Education, Health, and Recreation, National Chia-Yi University, Chia-Yi, Taiwan
[6] Department of Exercise and Health Promotion, Chinese Culture University, Taipei, Taiwan
[*] These authors contributed equally to this work.

Corresponding author
Frank Jing-Horng Lu,
frankjlu@gmail.com

## ABSTRACT

**Background.** Although Perceived Stress Scale (PSS, *Cohen, Kamarack & Mermelstein, 1983*) has been validated and widely used in many domains, there is still no validation in sports by comparing athletes and non-athletes and examining related psychometric indices.

**Purpose.** The purpose of this study was to examine the measurement invariance of PSS between athletes and non-athletes, and examine construct validity and reliability in the sports contexts.

**Methods.** Study 1 sampled 359 college student-athletes (males = 233; females = 126) and 242 non-athletes (males = 124; females = 118) and examined factorial structure, measurement invariance and internal consistency. Study 2 sampled 196 student-athletes (males = 139, females = 57, $M_{age}$ = 19.88 yrs, SD = 1.35) and examined discriminant validity and convergent validity of PSS. Study 3 sampled 37 student-athletes to assess test-retest reliability of PSS.

**Results.** Results found that 2-factor PSS-10 fitted the model the best and had appropriate reliability. Also, there was a measurement invariance between athletes and non-athletes; and PSS positively correlated with athletic burnout and life stress but negatively correlated with coping efficacy provided evidence of discriminant validity and convergent validity. Further, the test-retest reliability for PSS subscales was significant ($r = .66$ and $r = .50$).

**Discussion.** It is suggested that 2-factor PSS-10 can be a useful tool in assessing perceived stress either in sports or non-sports settings. We suggest future study may use 2-factor PSS-10 in examining the effects of stress on the athletic injury, burnout, and psychiatry disorders.

## INTRODUCTION

Since the development of the Perceived Stress Scale (PSS, *Cohen, Kamarack & Mermelstein, 1983*) it has been widely used in various research such as the degree of global stress of a given situation (*Leon et al., 2007*; *McAlonan et al., 2007*), or effectiveness of an intervention on psychological stress (*Holzel et al., 2010*; *Seskevich & Pieper, 2007*; *Taylor-Piliae et al., 2006*), or the associations of perceived stress and psychiatric/physical disorder (*Culhane et al., 2001*; *Garg et al., 2001*). In addition, many studies used PSS to examine its relationship with quality of life (*Golden-Kreutz et al., 2004*; *Golden-Kreutz et al., 2005*), job satisfaction (*Norvell et al., 1993*), immune functioning (*Burns et al., 2002*; *Maes & Van Bockstaele, 1999*), depression (*Carpenter et al., 2004*), and sleep quality (*Cohen & Williams, 1988*). Therefore, it can be said PSS is a very important tool in assessing stress.

Built on *Lazarus & Flokman*'s *(1984)* transactional model of stress, the development of PSS is to assess one's perceived nonspecific stress in a given situation or a daily life situation. *Lazarus & Flokman*'s *(1984)* transactional model of stress contends that an individual's stress perception derived from the imbalance between one's appraisal of situational demands and coping resources. If one perceives that situational demands over resources and the consequences of such failure will be severe; then, it will lead to psychophysiological responses such as fast heartbeats, pale face, cold and sweaty hands, tense muscles, and so on. Cohen and colleagues *(1983)*; *Cohen & Williams, 1988*) constructed this global stress perception by two important components—something that one can control (i.e., counter stress) and something that one can't control (i.e., perceived stress). In such manner, PSS is not only a measure to assess an extant of how a given situation might hurt oneself but also to assess the degree of how this give situation is controllable or uncontrollable (*Golden-Kreutz et al., 2004*; *Örücü & Demir, 2009*; *Roberti, Harrington & Storch, 2006*).

Given that the applicability of PSS in assessing perceived stress in many domains, the number of the item, the factorial structure, and the reliability of PSS have been intensively examined by many researchers. For example, although the initial version of PSS (*Cohen, Kamarack & Mermelstein, 1983*) has been developed as 14-item with a unidimensional measure, Cohen and colleagues *(1988)* continuingly examined the appropriateness of the item's number and factorial structure of PSS. They sampled 960 male and 1,427 female US residents ($M_{age} = 42.8 \pm 17.2$ years) and examined its factorial structure, criterion validity, and internal consistency of the PSS. Results found 10-item, by deleting item 4, 5, 12, and 13, 2-factor PSS-10 can be a better measuring tool of perceived stress because they found the revised version of PSS-10 accounted 48.9% of the variance, and had better reliabilities (Cronbach's $\alpha = .84 \sim .86$), and correlated with anxiety, depression and life events which indicated good construct validity.

Followed Cohen and colleagues *(1988)*, Hewitt and colleagues *(1992)* sampled psychiatry patient as participants and examined factorial structure and reliability of PSS, the exploratory factor analysis (EFA) found PSS had two factors—perceived distress (Cronbach's $\alpha = .81$) and perceived coping (Cronbach's $\alpha = .72$) with 11 items. Similar findings also found in a Mexico sample where *Ramírez & Hernández (2007)* found two factors PSS-10 had better reliability and accounted 48.02% of the variance. Recently,

Barbosa-Leiker and colleagues (*2013*) examined measurement invariance of PSS across gender and time with a clinical sample. Results indicated the 2-factor 10-PSS model provided acceptable fit in both men and women at each time point.

Based on above literature, we understand that earlier version of one-factor 14-PSS (*Cohen, Kamarack & Mermelstein, 1983*) was designed to measure a global perceived stress without considering the dimensionality. However, when researchers continuingly examined the psychometric properties of PSS (e.g., *Barbosa-Leiker et al., 2013*; *Cohen & Williams, 1988*; *Hewitt, Flett & Mosher, 1992*; *Ramírez & Hernández, 2007*) it has found that the items, the factorial structure and the reliability of PSS need further examination. Especially, researchers concerned about whether 2-factor 10-PSS or 14-PSS can be an ideal tool in assessing perceived stress. Further, since PSS was to assess individuals perceive normal stress in a given situation or a daily life situation, however, the stress of an athlete on the training and competing situation is different from non-athletes. We doubt whether PSS is suitable to be used in sports and the measurement is invariant comparing to non-athletes. Therefore, it is imperative to examine the psychometric properties of PSS in sports and compare the measurement invariance between athletes and non-athletes.

In sports, stress is an important issue that has been received much of attention; specifically in the studies of athletic burnout and athletic injury. For example, *Smith (1986)* proposed a cognitive-affective model of athletic burnout to explain the influences of stress on burnout. *Smith (1986)* explains that in the stressful sports settings, athletes keep appraising contextual stressors and personal coping resources. If athletes perceived situational demands surpass personal resources and consequences will be severe, the cognitive appraisal leads to severe physiological and psychological responses—anxiety, tension, insomnia, and illness, which eventually lead to burnout. With the same line of conceptualization, *Andersen & Williams (1988)* also proposed a 'stress-athletic injury model' which contends that athletic injury is the interaction between personality, history of stressors, coping resources and cognitive appraisal. In the stress appraisal process, athletes' perceived stress is influenced by above mentioned factors such as personality, history of stressors, and coping resources. The consequences of this interaction can lead to either attenuate or deteriorate the perceived stress and eventually cause athletic injury.

Many researchers borrow either stress-burnout or stress-injury concepts to examine the role of perceived stress on athletes' burnout and injury. Researchers use different measures to do their studies. For example, *Rushall (1990)* used Daily Analysis of Life Demands for Athletes (DALDA) to measure athletes' perceived environment stimuli and overtraining. *Petrie (1992)* used the Life Event Scale for Collegiate Athletes (LESCA) to measure athlete's life stressors and its relations to sports injury. Lu and his colleagues (*2016*) used College Student-Athlete Life Stress Scale (CSALSS) to examine the relationships between coaches' social support, athletes' resilience and burnout. The general problem of these measures is they are only used for assessing life stressors and examining its' relationship with athlete burnout or sports injury. Although some report adequate psychometric properties (e.g., *Lu et al., 2012*), but some have been questioned regarding its reliability and validity (e.g., DALDA, LESCA). If they are used for assessing global perceived stress they are not appropriate.

To assess global perceived stress in sports, some researchers used PSS in examining its' relationship with sports injury and burnout. For example, in examining whether stress and affect as the mediator of hope-burnout relationship, Gustafsson and colleagues *(2013)* administered 238 Swedish soccer players with trait hope, Swedish version of PSS (i.e., PSS-10), positive and negative affect and athlete burnout scales. Results found athletes' hope and burnout were fully mediated by stress and positive affect. Similarly, *Tashman, Tenenbaum & Eklund (2010)* sampled 177 college coaches and examined the relationship between coaches' perfectionism and burnout. Results indicated that perceived stress (measured by PSS-14) mediated the relationship between self-evaluative perfectionism and burnout, and a significant direct link to burnout, accounting for 56% of its variance. Similar sport burnout studies that using either PSS-10 or PSS-14 can be found in *Raedeke & Smith (2004)*, *Smith, Gustafsson & Hassmeín (2010)*, and *Gustafsson & Skoog (2012)* studies.

In terms of stress- athletic injury relationship, Galambos and colleagues *(2005)* investigated 845 Australian youth athletes' injury rates and its relations with psychological variables. Participants (males = 433; females = 412) completed a demographic questionnaire, health history, Brunel Mood Scale and PSS-10. Results found mood and stress collectively predicted injury characteristics. In a similar study, *Malinauskas (2010)* sampled 123 college athletes and administered with social support scale, PSS-10, and life satisfaction scale. Results found greater perceived stress was associated with diminished life satisfaction for major injury athletes than minor injury athletes. Also, the interaction between perceived stress and perceived social support was associated the most with diminished life satisfaction for athletes with a major injury.

As previously mentioned, although sports researchers used either PSS-10 or PSS-14 in examining their relationships with athletic burnout or injury, the psychometric properties of PSS have never been examined in sports. Thus, there are several questions remained. First, whether two-factor or unidimensional PSS will be suitable for sports? Second, if we compare athletes and non-athletes will measurement characteristics remains the same? Third, what is the reliability and validity of PSS in sports? Therefore, the existing knowledge gap of the application of PSS in sports is that we don't know which type of PSS is suitable in terms of dimensionality, construct validity, reliability, and group comparisons between athletes and non-athletes. To fill the gap, there are three purposes in this study. First, we intended to examine the factorial structure of the PSS-10 and PSS-14 and internal consistency. Second, we intended to examine the measurement invariance of PSS between athletes and non-athletes. Third, we attempted to examine the construct validity of PSS and test-retest reliability of PSS.

## STUDY 1

The purpose of study1 was threefold: (a) to compare the factorial structure of the PSS-10 and PSS-14; (b) to examine internal consistency of PSS-10 and PSS-14; (c) to examine measurement invariance of PSS between athletes and non-athletes.

 

## Methods

### Participants & procedure

We recruited 359 college student-athletes (males = 233; females = 126) with mean age 20.08 (SD = + 1.51) from two sport-colleges and three universities in Taiwan. At the time of data collection, athletic participants were all in their regular training seasons and had been participating in a variety of individual and team sports, such as gymnastics, track and field, golf, weightlifting, basketball, volleyball, soccer, Tae-kwon-do, badminton and baseball for 8.93 years (SD = 3.14) of training and competition experiences.

For non-athlete participants we recruited 242 (males = 124; females = 118; $M_{age}$ = 20.10 yrs, SD = + 1.55) participated in our study. We collected their data during their classes. They are all regular students studied at different academic departments such as accounting, political science, biology, mathematics...etc. They participated in our study voluntarily without any conditional requirements from the classes.

After the approval by a local institutional review board (Antai- Tian-Sheng memorial Hospital Institutional Review Board, TSMH IRB No. 15-055-B1), the first author contacted the coach/teacher of a target team and class and asked permission to use his/her team/class as participants. Once the coach/teacher agreed to use his/her team/class as participants, we visited target team/class one hour before they finished the regular training/class. Before administering the questionnaire package, the second author explained the general purpose of the study, the method to complete questionnaires and rights of being a participant. To prevent social desirability effects, we informed participants that this is a study to explore college students' life experiences, and there were no right or wrong answers. Additionally, we asked them to answer the questions as truthfully as possible, and all responses would be confidential. After the briefing, participants who interested in this study then signed a consent form and completed the demographic questionnaire and 14-item PSS. It took about 15 min to complete the questionnaires.

### Measurements

#### Demographic information

The demographic questionnaire was designed to gather information about participants' age, gender, types of sports, and years of athletic experiences.

#### Perceived Stress Scale (PSS)

The PSS is a self-report measure designed to assess one's perception about the degree of a given situation in daily life is considered stressful (*Cohen, Kamarack & Mermelstein, 1983*). The PSS-14 contains seven positively worded 'stress' items (e.g., How often have you felt upset because of something that happened unexpectedly?) and seven negatively worded 'counter-stress' items (e.g., How often have you felt confident about your ability to handle personal problems?). Items are rated on a 5-point Likert scale of occurrence these statements over the past 4 weeks (0 = never, 1 = almost never, 2 = sometimes, 3 = fairly often, 4 = very often). Because reverse-coding may confound the counter-stress factor (*Golden-Kreutz et al., 2004*), we did not reverse-code the items.

**Table 1   Fit indices for one-factor and two-factor models of the PSS.**

|  | *df* | $\chi^2$ | CFI | RMSEA | SRMR |
|---|---|---|---|---|---|
| **1-factor** |  |  |  |  |  |
| PSS14 | 77 | 1171.113[*] | 0.495 | 0.154 | 0.162 |
| PSS10 | 35 | 486.814[*] | 0.693 | 0.147 | 0.126 |
| **2-factor** |  |  |  |  |  |
| PSS14 | 76 | 361.388[*] | 0.868 | 0.079 | 0.078 |
| PSS10 | 34 | 100.712[*] | 0.955 | 0.057 | 0.049 |

**Notes.**
[*]$p < .001$.

## Analytic strategy

All primary statistical testing was conducted in AMOS version 22. Models of the PSS-14 and the PSS-10 (i.e., by deleting item 4, 5, 12, and 13) were first estimated separately for the 1-factor and 2-factor as Table 1. Overall, model fit was evaluated using the following indices suggested by *Hu & Bentler (1999)* as follow: the comparative fit index (CFI; study criterion $\geq 0.950$ as ideal and $\geq 0.90$ as the minimum acceptable level), the root mean square error of approximation (RMSEA; study criterion $\leq 0.080$) and the standardized root mean square residual (SRMR; study criterion $\leq 0.080$). To examine the internal consistency of the factors, Cronbach's $\alpha$ coefficient was used as an index.

For testing measurement invariance, we adopted earlier suggestion (*Barbosa-Leiker et al., 2011*) by following procedures: (a) once the confirmatory factor models for each group established that the overall model was acceptable, a series of analyses to examine measurement invariance were performed sequentially between comparison and nested model; (b) each model was added equality constraints and was tested against the less-constrained model including following indices:

1. Configural invariance (*Horn & McArdle, 1992*) (also referred to as 'equal form'). This step examined the pattern of salient and non-salient loadings across groups (*Vandenberg & Lance, 2000*). This step took the measurement model and examined if the theoretical framework of the PSS is the same for athletes and non-athletes.

2. Metric invariance (*Horn & McArdle, 1992*) (also referred to as 'equal loadings'). This step constrained the factor loadings for like items across groups to determine whether the expected changes in observed values of the indicators per unit change of the construct were equal (*Vandenberg & Lance, 2000*). This step tested if the relationships of the PSS-14 or PSS-10 items were equivalent for like indicators in athletes and non-athletes.

3. Factor variance/covariance invariance (also referred to as 'equal factor variances'). This step constrained the like factor variances across the groups (*Vandenberg & Lance, 2000*). If factor variance invariance holds, then the amount of within group variability of the latent factor is equal across groups (*Brown, 2006*). This step tested whether athletes and non-athletes use equivalent ranges of the latent constructs (stress and counter-stress) to respond to the PSS-14 or PSS-10 items.

**Table 2** Invariance models of the two-factor PSS-10 between athletes and non-athletes.

| | | df | $\chi^2$ | CFI | RMSEA | SRMR | ΔCFI |
|---|---|---|---|---|---|---|---|
| Athlete v.s. non-athlete | M1(configural) | 68 | 178.961[*] | 0.927 | 0.052 | 0.059 | |
| | M2(metric) | 76 | 193.196[*] | 0.923 | 0.051 | 0.061 | −0.004 |
| | M3(variance\ covariance) | 79 | 196.174[*] | 0.923 | 0.050 | 0.062 | 0.000 |
| | M4(residual) | 89 | 265.011[*] | 0.884 | 0.057 | 0.061 | −0.039 |

**Notes.**
[*]$p < .001$.

4. Error variance invariance. This step constrained error variances across the groups. If the same level of measurement error is present for each item between groups. This step tested whether the error of stress and counter-stress all items were related equivalently across athletes and non-athletes.

For tests of invariance, $\chi^2$ difference tests are typically used to compare nested models. However, the $\chi^2$ difference test may also be influenced by sample size (*Chen, Sousa & West, 2005*); thus, a change in the comparative fit index (CFI) between comparison and nested models of greater than or equal to −0.010. In addition, we examined the change in root mean square error of approximation (RMSEA) ≥0.015 or a change in standardized root mean square residual (SRMR) ≥0.030 (for loading invariance) and ≥0.010 (for intercept invariance) is recommended as an appropriate criterion indicating a decrement in fit between models (*Chen, 2007*; *Chen, Sousa & West, 2005*; *Cheung & Rensvold, 2002*). Additionally, a $\chi^2$ difference test for a small difference between models (rather than 0) was also conducted ($\chi^2_{\text{critical } 0.05}$; *MacCallum, Browne & Cai, 2006*).

## Results

Table 1 compares the fit of the 1-factor and 2-factor model of PSS-10 and PSS-14. Both 1-factor model of 14-items (RMSEA = 0.156 > 0.080; CFI = 0.483 < 0.90; SRMR = 0.157 > 0.080) and 10-items (RMSEA = 0.146 > 0.080; CFI = 0.685 < 0.90; SRMR = 0.146 > 0.080) did not fit very well. Only 2-factor PSS-10 (RMSEA = 0.070 < 0.080; CFI = 0.929 > 0.90; SRMR = 0.060 < 0.080) was the best model. Also, it was found Cronbach's $\alpha$ coefficients for 2-factor PSS-10 were .81 (perceived stress) and .71 (counter stress).

Table 2 shows the athletes and non-athletes 2-factor PSS-10 model of the measurement invariance, M1 was configuration invariance model, M2 metric invariance, M3 variation \ covariance invariance, M4 error variance invariance is shown to have an acceptable adaptation indicators. ΔCFI indicated that 2-factor PSS-10 model of athletes and non-athletes in M1, M2, M3 measurement invariance model display equivalent (ΔCFI < 0.01), however, M4 shows the residuals are not equal (ΔCFI > 0.01). We will discuss this later in the discussion.

## STUDY 2

The purpose of study 2 was to examine the construct validity of 2- factor PSS-10, which is the convergent and discriminant validity, via correlational analyses surrounding the relationships among PSS-stress, PSS counter-stress, college student-athletes' life stress, coping self-efficacy and burnout experiences.

## Methods

### Participants & procedure

A new sample of the targeted population was recruited. Valid data of 196 student-athletes from ten different universities were collected (males = 139, females = 57, $M_{age}$ = 19.88 yrs, $SD$ = 1.35). The recruiting procedure was similar to study 1.

### Measurements

The measurements included the Demographic Questionnaire and the 10-item PSS. In addition, the researchers administered the following measures for examining convergent and discriminant validity.

### Athlete burnout (ABQ)

ABQ (*Raedeke & Smith, 2001*) is a self-reported inventory that assesses athletes' burnout experiences. The initial factor analyses by *Raedeke & Smith (2001)* revealed that ABQ has three subscales including (a) five items for reduced sense of athletic accomplishment, (b) five items for perceived emotional and physical exhaustion, and, (c) 5 item for the devaluation of sports participation. Participants identify their athletic burnout experiences using a six-point Likert scale that ranged from 1 (*never*) to 6 (*always*). In the present study, the result of CFA confirmed that the factorial structure was suitable for the data. The Cronbach's $\alpha$ for the three subscales ranged from .63 to .86 and the reliability for all items was .90. To further identify convergent validity it is expected that the burnout scale should be positively correlated with the PSS stress because athletes' stress has been identified as a leading factor of athlete burnout (*Lewis, 1991*; *Nicholls et al., 2009*; *Galambos et al., 2005*; *Johnson & Ivarsson, 2011*). The PSS counter-stress factor was expected to have a negative relation with ABQ.

### College atudent-athletes' life stress scale (CSALSS)

The 24-item CSALSS (*Lu et al., 2012*) was used to assess situations that athletes encountered in their daily life and sports and considered as major stressors in their lives. The questionnaire asked questions such as "I am annoyed with my coach's bias against me." There are eight factors in the 24-item CSALSS including: (a) sports injury, (b) performance demand, (c) coach relationships, (d) training adaptation, (e) interpersonal relationships, (f) romantic relationships, (g) family relationships, and (h) academic requirements. Lu and colleagues *(2012)* reported that CSALSS can be categorized into two major components— general life stressors (by adding factor e, f, g, h) and sport-specific stressors (by adding factor a, b, c, d). Participants indicated the frequency of the event on a six-point Likert scale ranged from 1 (*Never*) to 6 (*Always*). Cronbach's $\alpha$ of these factors ranged from .69 to .87 and the reliability for all items was .92 in this study, indicating that the result was reliable. Given that CSALSS represents an individual's life stress, the PSS counter-stress factor was expected to have negative with CSALSS, while PSS stress was expected to have a positive relation with CSALSS.

### Coping self-efficacy scale (CSE)

CSE (*Chesney et al., 2006*) is a self-reported inventory that assesses one's confidence in performing coping behaviors when faced with life challenges. The initial factor analyses by

*Chesney et al. (2006)* revealed that CSE has three subscales including (a) problem- focused coping (6 items), (b) stop unpleasant emotions and thoughts (four items), and (c) get support from friends and family (three items). The CSE uses an eleven-point Likert scale that ranged from 0 ('can not do at all'), 5 ('moderately certain can do') to 10 ('certain can do'). In the present study, the Cronbach's $\alpha$ for the three subscales ranged from .70 to .78 and the reliability for all items was .86. It is expected that CSE will be negatively correlated with the PSS stress but positively correlated with PSS counter-stress.

## Statistical analyses

To examine the factor structure of the two factors PSS-10, maximum likelihood CFA using AMOS 22.0 was performed, and the researchers reported the following absolute and incremental fit indices if the 2-factors measurement model fit the data well: (1) the $\chi^2$/DF ratio; (2) the root mean square error of approximation (RMSEA); (3) the standardized root mean square residual (SRMR); (4) the Goodness of Fit Index (GFI); (5) the Comparative Fit Index (CFI); and (6) the Non-Normed Fit Index (NNFI). The recommendations for good fit are the following: $\chi^2$/DF ratios between one and three, values for RMSEA values less than 0.08 along with SRMR values less than 0.05, and GFI/CFI/NNFI values greater than 0.90 (*McDonald & Ho, 2002*; *Hu & Bentler, 1999*). To examine the internal consistency of the factors, Cronbach's $\alpha$ coefficient was used as an index. Additionally, the composite reliability (CR > .7) (*Fornell & Larcker, 1981*) and average variance extracted (AVE > .5) (*Kline, 1998*) were calculated to examine the fit of internal structure. Finally, using SPSS 18.0, the investigators examined the concurrent and discriminant validity by examining the correlations among PSS, CSALSS, CSE and ABQ.

## Results

The results of factorial structure suggest that 2-factor PSS-10 has better measuring quality than unidimensional PSS-10 or PSS-14. The two factors measurement model of the PSS-10 indicated a good fit of the instrument according to the fit indices in study2 (RMSEA = .056, SRMR = .061, $\chi^2$/DF = 1.607, CFI = .960, NNFI = .903, GFI = .955). The factor loadings for the 10 items range from .33 to .87 (Fig. 1). Cronbach's $\alpha$ coefficients of stress was .77, the coefficient for the "counter-stress" was .68. The composite reliability (*Fornell & Larcker, 1981*) for each subscale was calculated: stress (.78), and counter-stress (.73) indicating that each was above the .70 standard. The average variance extracted was also calculated: stress (.59), and counter-stress (.52), which were all above the acceptable standard (.50). As for concurrent and discriminant validity, the Pearson correlations indicated that PSS-stress negatively correlated with CSE total scores and CSE subscales, and positively related to ABQ total scores, ABQ subscales, CASLSS total scores, and CASLSS subscales. Moreover, PSS-counter stress negatively or not correlated with ABQ total scores, ABQ subscales, CASLSS total scores, and CASLSS subscales, in contrast, positively related with CSE (see Table 3).

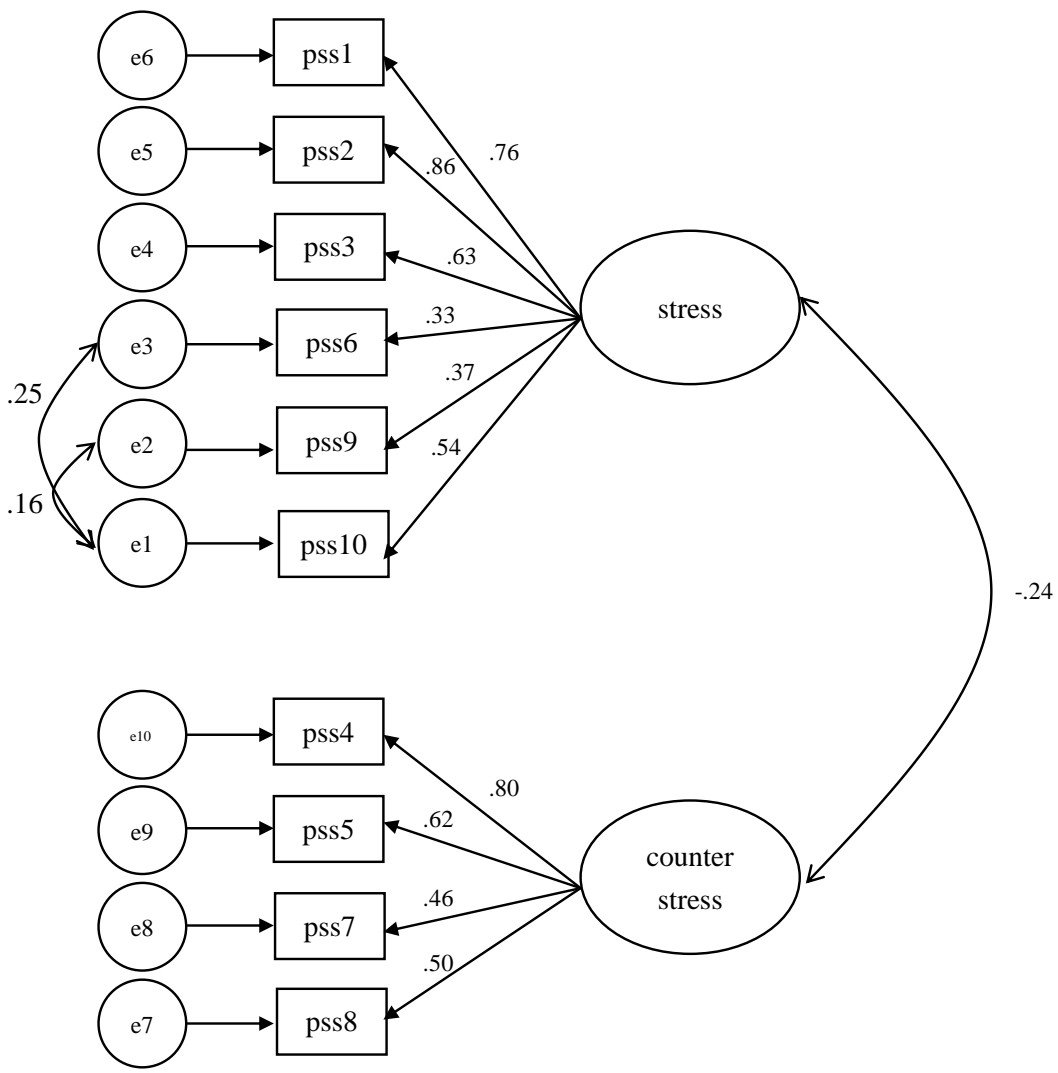

**Figure 1  The two factors measurement model of the PSS-10.**

## STUDY 3

Previous studies had found that the retest reliability was a better index of the reliability than the internal consistency coefficients (*McCrae, 2014*; *McCrae et al., 2011*). Therefore, the purpose of study 3 was to examine the test and retest reliability of 2- factor PSS-10.

### Methods
#### Participants & procedure
Another new sample was recruited from Judo and Boxing ($n = 39$). The data was collected in 8–9 days, and with the general procedures in the previous section. However, because two participants were unable to complete all items, their data were dropped from the study.

#### Measurements
The measurements included the Demographic Questionnaire and the 2-factor PSS-10.

**Table 3  The bivariate correlations of PSS, ABQ, CSE, and CSALSS.**

| | 1 | 2 | 3 | 4 | 5 | 6 | 7 | 8 | 9 | 10 | 11 | 12 | 13 | 14 | 15 | 16 | 17 | 18 | 19 |
|---|---|---|---|---|---|---|---|---|---|---|---|---|---|---|---|---|---|---|---|
| 1.Stress | .77 | | | | | | | | | | | | | | | | | | |
| 2.Counter—stress | −.13 | .68 | | | | | | | | | | | | | | | | | |
| 3.ABQ | .37* | −.29* | .90 | | | | | | | | | | | | | | | | |
| 4.CSE | −.29* | .41* | −.30* | .86 | | | | | | | | | | | | | | | |
| 5.CASLSS | .45* | −.18* | .46* | −.29* | .92 | | | | | | | | | | | | | | |
| 6.ABQ-RA | .31* | −.44* | .82* | −.31* | .36* | .63 | | | | | | | | | | | | | |
| 7.ABQ-E | .34* | −.17* | .86* | −.26* | .39* | .54* | .85 | | | | | | | | | | | | |
| 8.ABQ-D | .31* | −.18* | .90* | −.23* | .44* | .65* | .64* | .86 | | | | | | | | | | | |
| 9.CSE-PC | −.15* | .27* | −.19* | .74* | −.19* | −.28* | −.11 | −.14 | .76 | | | | | | | | | | |
| 10.CSE-SET | −.32* | .44* | −.29* | .92* | −.31* | −.32* | −.25* | −.21* | .52* | .78 | | | | | | | | | |
| 11.CSE-GF | −.20* | .28* | −.26* | .85* | −.22* | −.17* | −.26* | −.22* | .47* | .69* | .70 | | | | | | | | |
| 12.CASLSS-SJ | .31* | −.24* | .26* | −.24* | .68* | .21* | .27* | .20* | −.16* | −.23* | −.19* | .87 | | | | | | | |
| 13.CSALSS-PD | .46* | −.24* | .30* | −.34* | .65* | .32* | .28* | .21* | −.19* | −.37* | −.25* | .56* | .69 | | | | | | |
| 14.CSALSS-CR | .37* | −.08 | .46* | −.21* | .70* | .39* | .32* | .47* | −.13 | −.21* | −.19* | .38* | .39* | .84 | | | | | |
| 15.CSALSS-TA | .40* | −.15* | .57* | −.26* | .80* | .39* | .50* | .55* | −.10 | −.30* | −.23* | .45* | .45* | .64* | .70 | | | | |
| 16.CSALSS-IR | .17* | −.06 | .37* | −.16* | .69* | .28* | .25* | .42* | −.15* | −.14* | −.12 | .24* | .24* | .51* | .58* | .81 | | | |
| 17.CSALSS-RR | .19* | −.03 | .20* | −.05 | .70* | .14* | .17* | .20* | −.03 | −.09 | .01 | .31* | .24* | .31* | .52* | .52* | .75 | | |
| 18.CSALSS-FR | .28* | −.08 | .23* | −.12 | .79* | .20* | .15* | .24* | −.13 | −.13 | −.06 | .42* | .39* | .44* | .53* | .60* | .61* | .70 | |
| 19.CSALSS-AR | .41* | −.17* | .28* | −.29* | .74* | .18* | .28* | .25* | −.18* | −.32* | −.21* | .45* | .50* | .35* | .48* | .37* | .49* | .56* | .77 |
| *Mean* | 2.01 | 2.03 | 3.03 | 6.65 | 2.82 | 3.14 | 3.22 | 2.73 | 6.83 | 6.38 | 6.84 | 3.45 | 3.51 | 2.65 | 2.73 | 2.08 | 2.38 | 2.61 | 3.18 |
| *SD* | 0.67 | 0.70 | 0.85 | 1.37 | 0.79 | 0.80 | 1.03 | 1.13 | 1.88 | 1.61 | 1.46 | 1.29 | 0.94 | 1.19 | 1.00 | 0.97 | 1.15 | 1.08 | 1.21 |

**Notes.**

*$p < .05$; Cronbach alphas are presented on the diagonal as bold font.

ABQ-RA, reduced sense of athletic accomplishment; ABQ-E, perceived emotional and physical exhaustion; ABQ-D, devaluation of sports participation; CSE-PC, problem-focused coping; CSE-SET, stop unpleasant emotions and thoughts; CSE-GF, get support from friends and family. CSALSS included in eight factors; SJ, sports injury; PD, performance demand; CR, coach relationships; TA, training adaption; IR, interpersonal relationships; RR, romantic relationships; FR, family relationships; AR, academic requirements.

## Statistical analyses

We used SPSS 18.0 to analyze the raw data. The Pearson correlations were used to assess the test-retest reliability.

## Results

Results indicated that the Pearson coefficients for perceived stress ($r = .66, p < .00$), and counter stress was ($r = .50, p < .00$) which indicated that two subscales significantly reliable over measuring time.

## DISCUSSION

In line with past research examining the validity of PSS the purpose of this study was to examine psychometric properties of PSS in sports settings. Specifically, this study attempted to examine factorial structure, measurement invariance, reliability, and construct validity of PSS. By three studies we found 2-factor PSS-10 has a better fit of the model. Also, we found 2-factor PSS-10 had appropriate internal consistency, testretest reliability and measurement invariance across athletes and non-athletes. Further, we found 2-factor PSS-10 positively correlated with athletes' life stress and burnout but negatively correlated with coping self-efficacy which indicated appropriate construct validity.

Therefore, the psychometric properties of PSS gain solid supports in the sports contexts. The results of factorial structure suggest that 2-factor PSS-10 has better measuring quality than 2-factor PSS-14, or 1-factor PSS-10/PSS-14. The 2-factor PSS-10 reduces four items allows researchers collect data in a short period of time (*Shacham, 1983*). Although researchers have different arguments regarding dimensionality of PSS (*Hewitt, Flett & Mosher, 1992*; *Mitchell, Crane & Kim, 2008*; *Örücü & Demir, 2009*) it is for sure that 2-factor PSS-10 receive better support in sports contexts. The results are consistent with the earlier study by Hewitt and colleagues *(1992)*. Past research in sports generally just assessed life stressors (e.g., CSALSS, DALDA, LESCA) or arbitrarily used PSS-14/PSS-10 in examining the relationship between stress and related psychological responses (e.g., *Raedeke & Smith, 2004*; *Smith, Gustafsson & Hassmeín, 2010*; *Gustafsson & Skoog, 2012*). They did not exactly understand the psychometric properties of PSS. Our results provide robust evidences that 2-factor PSS-10 can be an appropriate tool in the sports settings. Thus, researchers not only have a better tool in examining the relationship between perceived stress and related psychological responses but also practitioners can use 2-factor 10-PSS in evaluating athletes' existing perceived stress. Also, both researchers and practitioners can use two factors—perceived stress and counter stress to further understand what factor plays an important role on athletes' psychological responses. We suggest future researchers may use 2-factor PSS-10 to examine gender differences, or to compare which variable predict related psychological disorders/constructs the most.

In terms of measurement invariance, we found configuration invariance, metric invariance, and variance/covariance invariance are all equivalent except error variance invariance. Therefore, it means that the same level of measurement error for each item between athletes and non-athletes is not the same. However, *Lee (2006)* suggested that most research that using CFA focus on the equivalence of the factor loadings and factorial

covariance. If these indicators meet criteria they can assure that means measurement invariance across observed groups is held, while residual restrain model may be too critical to be reached. *Tabachnick & Fidell (2001)* also suggest that when factor loadings and factorial covariance are equivalent across group it is indicated that measurement invariance holds true.

A significant feature of this study is the sample recruited from Taiwanese student-athletes. Although PSS has been validated in different culture such as Spanish (*Remor, 2006*), Swedish (*Eskin & Parr, 1996*), Japanese (*Mimura & Griffiths, 2004*), Portugal (*Ramírez & Hernández, 2007*), Turkish (*Örücü & Demir, 2009*), or Chinese (*Leung, Lam & Chan, 2010*), this is the first study using healthy young athletes as participants in examining psychometric properties of PSS and its relationship with athlete burnout, coping self-efficacy and life stress. Our study not only confirms Leung and colleagues' *(2010)* psychometric properties of the Chinese version of PSS which indicated that 2-factor 14-PSS, 10-PSS and 4-PSS have better measurement quality but also extends the applicability of PSS into sports contexts. The results of psychometric validation and measurement invariance can be forwarded to those researchers who interested in psychometric properties of PSS. Further, our construct validity analyses found 2-factor PSS-10 correlated with athlete burnout and life stress, but negatively correlated with coping self-efficacy, are worthy of forwarding these messages to sports professionals. As we all know the young athletes face many challenges in their life (*Lewis, 1991*). They engage in heavy and intensive training/competition all year round. The intensive and heavy training bring lots of stress for young athletes (*Weinberg & Gould, 2015*). If coaches and sports professionals fail to monitor athletes' training loading and arrange them with appropriate competition plans they may induce lots of stress and cause burnout. Therefore, teaching young athletes effective coping skills (e.g., fostering social support and time management) and teaching psychological skills (e.g., goal-setting, relaxation, and imagery) are very important because they can help young athletes to cope with stress from training and competitions.

Further, as previously mentioned that most research in sports either using *Smith (1986)* cognitive-affective model of athletic burnout or *Andersen & Williams (1988)* stress-athletic injury model to examine the effects of stress on athlete burnout/injury, we suggest future research may extend research beyond this area. Especially, we suggest that future research may examine the antecedents of perceived stress in sports. For example, sports literature suggests that motivational climate created by coaches and teams (e.g., ego-involving climate) is one of the major sources of athletes' stress (*Hogue et al., 2013*). Researchers may use 2-factor PSS-10 in examining how a sports team's motivational climate predicts athletes' perceived stress. In addition, it is well-documented that the leadership style implemented by coaches (e.g., autocratic) may produce stress for athletes (*Horn et al., 2011*). We suggest researchers may examine how coaches' leadership and situational conditions (e.g., competition performance) predicting athletes' perceived stress. Further, since the athletic world is a challenging setting it is found that some athletes become substance abuse and eating disorder to cope stress (*Ansel, 2010*). We suggest researchers may use 2-factor PSS-10 as a measuring tool in assessing athletes' perceived stress during

off- season, pre-season, and after-season in order to understand their stress level and provided with appropriate interventions.

As to the application of 2-factor PSS-10 in other domains, we suggest it can be a useful tool in assessing perceived stress in an intervention study. For example, if researchers want to know whether a stress management program reducing participants' perceived stress (e.g., *Shapiro et al., 2005*). In such condition, researchers can use 2-factor PSS-10 before and after intervention so to understand the effectiveness of the intervention. Also, many researchers adopt a psychophysiological approach to examined participants' electroencephalography (EEG) during stressful conditions (e.g., *Cavanagh & Shackman, 2015*; *Grupe & Nitschke, 2013*; *Proudfit, Inzlicht & Mennin, 2013*). In such case, if researchers use 2-factor PSS-10 combined with EEG it can be a sound approach to detect participants' real psychological reactions. Further, many researchers in medicine intend to examine whether there is an association between perceived stress and coping strategies (e.g., *Najam & Aslam, 2010*). For this type of study 2-factor, PSS-10 can be a quick and efficient tool for collecting data.

There are several limitations that need to be addressed. First, our sample is all recruited from Division I college student-athletes whether our results could be generalized to other athletes, such as professional athletes or junior athletes, need to be further examined. Additionally, the data was collected from Taiwanese student-athletes; hence, the results may not be generalizable to different cultures. We recommend researchers adopt similar approaches to test measurement invariance in different cultures and populations.

## CONCLUSION

To acknowledge that PSS is a widely used measure in assessing stress, we have conducted three studies to examine the factor structure, measurement invariance between athletes and non-athletes, internal reliability, test-retest reliability and construct validity in the sports contexts. Results indicated that 2-factor PSS-10 can be an ideal measure for the research in sports. We suggest future research may use 2-factor PSS-10 in conducting various stress research.

### Funding

This study was partly supported by grants from Ministry of Science and Technology in Taiwan, MOST 104-2410-H-179-009. The funders had no role in study design, data collection and analysis, decision to publish, or preparation of the manuscript.

### Grant Disclosures

The following grant information was disclosed by the authors:
Ministry of Science and Technology: 104-2410-H-179-009.

### Competing Interests

The authors declare there are no competing interests.

## Author Contributions

- Yi-Hsiang Chiu performed the experiments, analyzed the data, wrote the paper, prepared figures and/or tables, check mechanical parts of the paper.
- Frank Jing-Horng Lu conceived and designed the experiments, contributed reagents/materials/analysis tools, wrote the paper, pay the necessary fees for the study.
- Ju-Han Lin performed the experiments.
- Chiao-Lin Nien reviewed drafts of the paper.
- Ya-Wen Hsu analyzed the data, contributed reagents/materials/analysis tools, reviewed drafts of the paper.
- Hong-Yu Liu performed the experiments.

## Human Ethics

The following information was supplied relating to ethical approvals (i.e., approving body and any reference numbers):

Antai Medical Care Cooperation Antai- Tian-Sheng memorial Hospital Institutional Review Board, TSMH IRB No. 15-055-B1.

## Data Availability

The raw data has been supplied as Supplementary Files.

## Supplemental Information

Supplemental information for this article can be found online at http://dx.doi.org/10.7717/peerj.2790#supplemental-information.

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
