# Peer review of "Psychometric properties of the Perceived Stress Scale (PSS): measurement invariance between athletes and non-athletes and construct validity"

_PeerJ, doi:10.7717/peerj.2790_

## Round 0.1 · original submission · Major Revisions

I now have received two reviewers' comments. Although both reviewers expressed their interest in your study, several aspects of this manuscript should be revised to improve its clarity. Especially reviewer 2 has pointed out several conceptual and methodological issues that require your attention. Their observations are presented with clarity so I'll not risk confusing matters by belaboring or reiterating their comments. While I might quibble with the occasional point, I note that I regard the reviewers' opinions as substantive and well-informed. I believe that all of the highlighted reservations require contemplation and appropriate attention in revising the document if it is to contribute appropriately to Peer J and the extant literature. Please revise or refute according to the two reviewers' comments and provide a point by point reply in addition to the revised manuscript.

Tsung-Min Hung, Ph.D.
PeerJ editor
Distinguished professor
Department of Physical Education
National Taiwan Normal University

Reviewer 1 ·

Basic reporting

1) Clear, unambiguous, professional English language used throughout.
Overall this study is clear, but some grammatical or structural written errors, and typo are needed to be revised. Please take time to proofread the entire manuscript. For example, Line 96-100, with same line of conceptualization, Andersen and Williams (1988) also proposed a ‘stress-athletic injury model’ which “stating” that athletic injury is the interaction between personality, history of stressors, coping resources and cognitive appraisal. In the stress appraisal process, athletes’ perceived stress is “influence” by above mentioned factors such as personality, history of stressors, and coping “rescources”.

2) Intro & background to show context.
The paper clearly introduces the development, applicability, validity of the different PSS versions. However, the authors are recommended to emphasize the meanings of applying the PSS to sports context, regardless of some instruments used in previous studies to measure stress. What are the existing knowledge gaps of research regarding the measurement of athletes’ stress perception? How can the PSS fill them?

3) Literature well referenced & relevant.
Literature has been well-referenced and relevant. In addition to the PSS, other inventories used to measure athletes’ stress are recommended to report and comment on their shortcomings.

4) Structure conforms to PeerJ standard, discipline norm, or improved for clarity.
The structure of the manuscript conforms to the standard of the PeerJ and the norm of relevant disciplines. As mentioned above, some arguments are needed to be further clarified.

5) Figures are relevant, high quality, well-labelled & described.
Tables are relevant that help readers realize the relationships among the variables, and the results of competing different models. For Table 3, a general note is required to provide explanations for some abbreviations. For the results of correlation, mean, and standard deviation, two decimal places are suggested. The format of table title is recommended to keep consistent and follow the standard of APA style. The authors should further refine the quality of the tables.

Experimental design

1) Original primary research within Scope of the journal.
This study is well-focused and properly builds on previous work. The purpose of this study falls within the scope of the journal, which is to examine the validity of the Perceived Stress Scale within the sport realm and investigate its associations with athlete burnout and student-athletes’ life stress.

2) Research question well defined, relevant & meaningful. It is stated how research fills an identified knowledge gap.
The description of research question is well-stated that makes the potential of this study to bring new relevant information to the field clear at this point. The authors are suggested to add some research regarding the measurement of athletes’ stress, and to further argue the existing knowledge gaps and then to point out how this study can solve these questions.

3) Rigorous investigation performed to a high technical & ethical standard.
This study is well designed and also follows the ethical standard to claim the approval of the IRB while recruiting athletes and non-athletes as the participants.

4) Methods described with sufficient detail & information to replicate.
The materials and methods described in the manuscript are moderately sufficient for other researchers to replicate. For Study 1, Line 148-150, demographic data for athletes and non-athletes are needed to be reported separately.

Validity of the findings

1) Negative/inconclusive results accepted.
In general, all relationships among the PSS, athlete burnout, and life stress were statistically significant, which were mostly consistent with the previous findings and the directions of the hypotheses.

2) Data is robust, statistically sound, & controlled.
The data provided is robust enough because this study utilized a two-stage design with different samples to examine the construct validity of the PSS via CFA, and to investigate the discriminant and convergent validity of the PSS. For the results of Study 1, Line 232-254, since all the statistical results are presented in Table 1 and Table 2, the authors are suggested to not replicate the description same with the tables and to give solid descriptions to inform the readers about the results. For Study 2, due to the multidimensional nature of measures used, such as the ABQ (3 subscales), the CSALSS (2 subscales), and the CSE (3 subscales), the relationships among these subscales and the 2 subscales of the PSS can give more information on the validity of the PSS. Line 336-337, the authors might make an incorrect description- The results of factorial structure suggest that 2-factor PSS-10 has better measuring quality than unidimensional “2-factor” PSS-10 or 2-factor PSS-14.

4) Conclusion well stated, linked to original research question & limited to supporting results.
The conclusion is appropriately stated based on the results of this study and the findings of previous research. Also, the authors have raised their arguments and recommendations according to the findings of this study. Regarding the better psychometric properties of the 2-factor PPS-10 within the sport context, the underlying theoretical basis of this model should be discussed in order to provide evidence on why the validity of this measurement model is superior than that of the other competing models when used to evaluate athletes’ stress.

Additional comments

Although some minor revisions are needed, generally, the justification of research questions is clear which makes this study contribute to the field regarding athletes’ stress. More, the method of this study is suitable and the validity and reliability of the measures are appropriate, which can increase the contribution of the findings. According to the findings, this study is able to provide new information regarding the measurement of athletes’ stress.

Reviewer 2 ·

Basic reporting

No Comments

Experimental design

No Comments

Validity of the findings

No Comments

Additional comments

The authors clearly define a research problem and topic on this article. Also, this article links theory and practice in an important way. The literature review is generally excellent but has several shortcomings that undermine its clarity and make the reader question the relevance of this study. The following comments are:

1.The introduction of the paper delves immediately into the specific research scenario. I would like to see the authors spend a paragraph to discuss what has motivated the research. Before getting into the conceptual model for the paper, I would like to see a clear statement of the purpose of this research. Especially, the rational of the difference between one order factor instrument and two order factor instrument.
2.The current study didn’t mention the reliability and validity of the Chinese perceived stress scale. So, how to confirm the Chinese perceived stress scale was equal to original scale?
3.The perceived stress scale has two important components (i.e., counter and perceived). Did the authors examine a second-order factor account for the relationships among the first-order factors?
4.The PSS was to assess individuals perceive normal stress in a given situation or a daily life situation, however, the stress of an athlete on the training and competing situation is different from non-athletes. In my opinion, it is better to provide a comprehensive and strong reasoning to examine the measurement invariance of PSS in athletes and non-athletes.
5.In this study, the researchers try to examine the measurement invariance (MI) of PSS. According to the results, the measurement invariance of PSS was provided. However, what’s the specific contribution in sporting context? The present study sought to explain the meaningful contribution to the literature.
6.Regarding to the reliability of PSS, please present the test retest reliability of the instrument (McCrae, 2014; McCrae, Kurtz, Yamagata, & Terracciano, 2011).
7.I would like to see much more discussion on what the findings mean to practitioners and researchers. What do the findings suggest we do differently? What are the implications of these findings?

---

## Round 0.2 · Minor Revisions

I have now received two reviewers’ comment and both reviewers were generally satisfied with your reply and revisions from previous comments. However, there are some issues require further revision. Therefore, I'd like to invite you to rewrite your manuscript according to the reviewers' comments. Please resubmit your revised manuscript along with a point by point reply to the reviewers' comments.

Tsung-Min Hung, Ph.D.
PeerJ editor
Distinguished professor
Department of Physical Education
National Taiwan Normal University

Reviewer 1 ·

Basic reporting

1) Clear, unambiguous, professional English language used throughout.
Overall this study is clear, but some grammatical or structural written errors, inappropriate punctuation marks, and typo are still needed to be revised to enhance the quality of this manuscript. Please take time to proofread the entire manuscript. At least, some inappropriate sentences are found as follows.

Line 109-113: However, when researchers continuingly examined the psychometric properties of PSS (e.g., Barbosa-Leiker, et al., 2013; Cohen & William, 1988; Hewitt, et al., 1992; Ramírez & Hernández, 2007) “the items, the factorial structure, and the reliability of PSS becoming a hot issue about PSS”. Especially, researchers concerned “bout” whether 2-factor 10-PSS or 14-PSS can be an ideal tool in assessing perceived stress.
Line 167-169: “As previously mentioned, although sports researchers used either PSS-10 or PSS-14 in examining their relationships with athletic burnout or injury. The psychometric properties of PSS have never been examined in sports, Thus, there are several questions remained.”
Line 195-196: They participated (in) our study voluntarily without any conditional requirements from the classes.
Line 214-215: The PSS ” measures” is a self-report measure designed to assess one’s perception about the degree of a given situation in daily life is considered stressful (Cohen et al., 1983).
Line 400: Past research in sports generally just “assessing” life stressors (e.g., CSALSS, DALDA, LESCA) or arbitrarily used PSS-14/PSS-10 in examining the relationship between stress and related psychological responses (e.g., Raedeke & Smith,2004; Smith, Gustafsson, and Hassmén,2010; Gustafsson & Skoog,2012).

2) Intro & background to show context.
In the revised manuscript, the authors have emphasized the existing knowledge gaps of research regarding the measurement of athletes’ stress perception. The importance of this study has been illustrated.

3) Literature well referenced & relevant.
Literature has been well-referenced and relevant.

4) Structure conforms to PeerJ standard, discipline norm, or improved for clarity.
The structure of the manuscript conforms to the standard of the PeerJ and the norm of relevant disciplines.

5) Figures are relevant, high quality, well-labelled & described.
Tables are relevant that help readers realize the relationships among the variables, and the results of competing different models. The titles of Table 1 and 2 are sort of inappropriate and recommended to be refined. For example, Table 1: The comparing table of PSS one-factor -model and two-factor- model. The title is suggested that “Fit indices for one-factor and two-factor models of the PSS.” Please attempt to give the tables brief but clear titles.

Experimental design

1) Original primary research within Scope of the journal.
The purpose of this study falls within the scope of the journal, which is to examine the validity of the Perceived Stress Scale within the sport realm and investigate its associations with athlete burnout and student-athletes’ life stress.

2) Research question well defined, relevant & meaningful. It is stated how research fills an identified knowledge gap.
The authors have added some information regarding the measurement of athletes’ stress, and further argued the existing knowledge gaps and how this study could solve these questions.

3) Rigorous investigation performed to a high technical & ethical standard.
This study is well designed and also follows the ethical standard to claim the approval of the IRB while recruiting athletes and non-athletes as the participants.

4) Methods described with sufficient detail & information to replicate.
The materials and methods described in the manuscript are moderately sufficient for other researchers to replicate.

Validity of the findings

1) Negative/inconclusive results accepted.
In general, all relationships among the PSS, athlete burnout, and life stress were statistically significant, which were mostly consistent with the previous findings.

2) Data is robust, statistically sound, & controlled.
The data provided is robust enough because this study utilized a three-stage design with different samples to examine the construct validity of the PSS via CFA, and to investigate the discriminant and convergent validity, and re-test reliability of the PSS.

4) Conclusion well stated, linked to original research question & limited to supporting results.
The conclusion is appropriately stated based on the results of this study and the findings of previous research.

Additional comments

The findings of this study contribute to the application of assessing athletes’ stress. Some minor revisions of writing are suggested for improving the quality of this manuscript.

Reviewer 2 ·

Basic reporting

No Comments

Experimental design

No Comments

Validity of the findings

No Comments

Additional comments

The authors have revised the article based on the previous review comments. First, this article conducted a cross-validation study for multi samples which is a popular strategy for model stability and validity extension. Second, the authors already reported the test retest reliability for the internal-consistency reliability. However, I suggest that authors add McCrae (2014) and McCrae, Kurtz, Yamagata, and Terracciano (2011) literature as citation during recent 5 years. Finally, I would like to see much more discussion or suggestion on the validity generalization for the psychometric properties of the perceived stress scale (PSS).

---

## Author Rebuttal · Round 0.2

Editor in Chief

Peer J

Dear professor.Tsung-Min Hung:

We would like to thank the journal offering us the opportunity to submit a revised version of our manuscript "Psychometric properties of the perceived stress scale (PSS): measurement invariance between athletes and non-athletes and construct validity". We also appreciate two reviewers for providing valuable and constructive comments and suggestions. We believe that their efforts, along with our revising, will make this paper better and better. Now, we followed all of your recommendations and addressed all the major concerns that were raised by reviewers. Moreover, before submitting a revised version, we used "Grammarly" software to check our spellings and grammar. Also, we check our references in the text twice. Now the quality of writing is guaranteed.

Again! We would like to express our sincere gratitude to you and the Editors-in-chief for providing these valuable recommendations and suggestions, which allowed us to improve the quality of this paper. Below, we explain, point by point, how we have incorporated the issues raised by the reviewers. To ease understanding, we have included the reviewer´s comments in italics followed by our own. Again! thank you for allowing us to resubmit a revised copy of the manuscript:

Reviewer 1

Basic reporting

1) *Clear, unambiguous, professional English language used throughout.*

   *Overall this study is clear, but some grammatical or structurally written errors, and typo are needed to be revised. Please take the time to proofread the entire manuscript. For example, Line 96-100, with the same line of conceptualization, Andersen and Williams (1988) also proposed a 'stress-athletic injury model' which "stating" that athletic injury is the interaction between personality, history of stressors, coping resources and cognitive appraisal. In the stress appraisal process, athletes' perceived stress is "influence" by above-mentioned factors such as personality, history of stressors, and coping "rescources".*

Reply: Thanks for the comments and suggestions. Indeed, there are several places have been wrongly typed and spelled. Now we revise them from line 86-95 and highlighted.

2) Intro & background to show context.

*The paper clearly introduces the development, applicability, validity of the different PSS versions. However, the authors are recommended to emphasize the meanings of applying the PSS to sports context, regardless of some instruments used in previous studies to measure stress. What are the existing knowledge gaps of research regarding the measurement of athletes' stress perception? How can the PSS fill them?*

Reply: Thank you for the valuable suggestions. We add a paragraph from line 156-159 do address

3) Literature well referenced & relevant.

*Literature has been well-referenced and relevant. In addition to the PSS, other inventories used to measure athletes' stress are recommended to report and comment on their shortcomings.*

Reply: Thank you for the comments and valuable suggestions. We add a paragraph to report other inventories used to measure athletes' perceived stress and comment on their shortcomings from line 111-124.

*4) Structure conforms to PeerJ standard, discipline norm, or improved for clarity. The structure of the manuscript conforms to the standard of the PeerJ and the norm of relevant disciplines. As mentioned above, some arguments are needed to be further clarified.*

Reply: Thank you for the comments! We add above paragraphs followed your suggestions.

*5) Figures are relevant, high quality, well-labelled & described. Tables are relevant that help readers realize the relationships among the variables*

*and the results of competing different models. For Table 3, a general note is required to provide explanations for some abbreviations. For the results of correlation, mean, and standard deviation, two decimal places are suggested. The format of table title is recommended to keep consistent and follow the standard of APA style. The authors should further refine the quality of the tables.*

Reply: Thank you for the valuable suggestions! On table 3, we add a general note to explain abbreviations. Also, for the results of correlation, mean, and standard deviation, we use two decimal places. Further, the format of table title has followed the standard of APA style. Please see revised table 3. Thank you!

Experimental design

*1) Original primary research within Scope of the journal.*

*This study is well-focused and properly builds on previous work. The purpose of this study falls within the scope of the journal, which is to examine the validity of the Perceived Stress Scale within the sports realm and investigate its associations with athlete burnout and student-athletes' life stress.*

Reply: Thank you for the comments!

*2) Research question well-defined, relevant & meaningful. It is stated how research fills an identified knowledge gap.*

*The description of the research question is well-stated that makes the potential of this study to bring new relevant information to the field clear at this point. The authors are suggested to add some research regarding the measurement of athletes' stress, and to further argue the existing knowledge gaps and then to point out how this study can solve these questions.*

Reply: Thank you for the valuable comments and suggestions! We have added some research regarding the measurement of athletes' stress on line 111-124. We further argue the existing knowledge gaps and point out how this study can solve these questions on line 156-159.

*3) Rigorous investigation performed to a high technical & ethical standard.*

*This study is well designed and also follows the ethical standard to claim the approval of the IRB while recruiting athletes and non-athletes as the participants.*

Reply: Thank you for the comments!

*4) Methods described with sufficient detail & information to replicate.*

*The materials and methods described in the manuscript are moderately sufficient for other researchers to replicate. For Study 1, Line 148-150, demographic data for athletes and non-athletes are needed to be reported separately.*

Reply: Thank you for the valuable comments and suggestions! For Study 1, we report separately on line 172-183 for athletes and non-athletes.

Validity of the findings

1) Negative/inconclusive results accepted.

*In general, all relationships among the PSS, athlete burnout, and life stress were statistically significant, which were mostly consistent with the previous findings and the directions of the hypotheses.*

Reply: Thank you for the comments!

*2) Data is robust, statistically sound, & controlled.*

*The data provided is robust enough because this study utilized a two-stage design with different samples to examine the construct validity of the PSS via CFA, and to investigate the discriminant and convergent validity of the PSS. For the results of Study 1, Line 232-254, since all the statistical results are presented in Table 1 and Table 2, the authors are suggested to not replicate the description same with the tables and to give solid descriptions to inform the readers about the results.*

Reply: Thank you for the valuable comments and suggestions! To avoid replicating the description of table 1 and table 2 tables and to give solid descriptions to inform

the readers we deleted several redundant sentences so make it solid and clear as line 261-267 indicated .

*For Study 2, due to the multidimensional nature of measures used, such as the ABQ (3 subscales), the CSALSS (2 subscales), and the CSE (3 subscales), the relationships among these subscales and the 2 subscales of the PSS can give more information on the validity of the PSS. Line 336-337, the authors might make an incorrect description- The results of factorial structure suggest that 2-factor PSS-10 has better-measuring quality than unidimensional "2-factor" PSS-10 or 2-factor PSS-14.*

Reply: Thank you for the comments and suggestions! Yes! It would be better to provide all measures' subscales and their correlations. We have added more information on table 2.   As to line 336-337, the correct description should be as "…the results of factorial structure suggest that 2-factor PSS-10 has better-measuring quality than 2-factor PSS-14 or 1-factor PSS-10/PSS-14" as line 353-354 indicated.

4) *Conclusion well stated, linked to original research question & limited to supporting results.*

*The conclusion is appropriately stated based on the results of this study and the findings of previous research. Also, the authors have raised their arguments and recommendations according to the findings of this study. Regarding the better psychometric properties of the 2-factor PPS-10 within the sports context, the underlying theoretical basis of this model should be discussed in order to provide evidence on why the validity of this measurement model is superior to that of the other competing models when used to evaluate athletes' stress.*

Reply: Thank you for the comments and suggestions! Yes, we have added a small paragraph to describe how the better psychometric properties of the 2-factor PPS-10 within the sports context, the underlying theoretical basis of this model is

superior to that of the other competing models in evaluating athletes' stress on line 408-419.

*Although some minor revisions are needed, generally, the justification of research questions is clear which makes this study contribute to the field regarding athletes' stress. More, the method of this study is suitable and the validity and reliability of the measures are appropriate, which can increase the contribution of the findings. According to the findings, this study is able to provide new information regarding the measurement of athletes' stress.*

Reply: Thank you for the comments and suggestions!

Reviewer 2 (Anonymous)

Basic reporting

No Comments

Experimental design

No Comments

Validity of the findings

No Comments

*Comments for the Author*

*The authors clearly define a research problem and topic in this article. Also, this article links theory and practice in an important way. The literature review is generally excellent but has several shortcomings that undermine its clarity and make the reader question the relevance of this study. The following comments are:*

1. *The introduction of the paper delves immediately into the specific research scenario. I would like to see the authors spend a paragraph to discuss what has motivated the research. Before getting into the conceptual model for the paper, I would like to see a clear statement of the purpose of this research. Especially,*

*the rational of the difference between one order factor instrument and two order factor instrument.*

Reply: Thank you for the valuable comments and constructive suggestions! We add a paragraph on line 86-95 to explain the difference of one-factor PSS and two-factor PSS, and why it is worthy of doing the research.

*2.The current study didn't mention the reliability and validity of the Chinese perceived stress scale. So, how to confirm the Chinese perceived stress scale was equal to original scale?*

Reply: Thank you for the valuable suggestions! Indeed, it is very informative to compare current study and Chinese PSS study. We add a small paragraph to discuss how our results mean to Cheung and colleagues' (2010) study on 438 and 441.

*3.The perceived stress scale has two important components (i.e., counter and perceived). Did the authors examine a second-order factor account for the relationships among the first-order factors?*

Reply: Thank you for the valuable comments! Yes, we tried to examine a first-order and second-order model of 2-factor PSS-10 but the only first-order model has been supported.

*4. The PSS was to assess individuals perceive normal stress in a given situation or a daily life situation, however, the stress of an athlete on the training and competing situation is different from non-athletes. In my opinion, it is better to provide a comprehensive and strong reasoning to examine the measurement invariance of PSS in athletes and non-athletes.*

Reply: This is a valuable suggestions! We have provided a paragraph to argue why comparing athletes and non-athletes is so important on line 86-95.

*5.In this study, the researchers try to examine the measurement invariance (MI) of PSS. According to the results, the measurement invariance of PSS was provided. However, what's the specific contribution in sporting context? The present study sought to explain the meaningful contribution to the literature.*

Reply: Thank you for the comments! The first reviewer also comments on the same question. To strengthen our rationale for the study, we explain the meaningful contribution of our study on line 156-159.

6.Regarding the reliability of PSS, please present the test-retest reliability of the instrument (McCrae, 2014; McCrae, Kurtz, Yamagata, & Terracciano, 2011).

Reply: We appreciate your valuable comment constructive suggestion! To establish a test-retest reliability we have conducted an additional study on study 3 on line 370-388. Thank you for the suggestion. We think this suggestion makes this paper more complete!

*7. I would like to see much more discussion on what the findings mean to practitioners and researchers. What do the findings suggest we do differently? What are the implications of these findings? What do the findings suggest we do differently? What are the implications of these findings?*

Reply: Thank you for the comments! To discuss what findings mean to practitioners and researchers, we add a paragraph to discuss this issue on line 408-419. Also, on line 447-455 we discuss the implications of these findings.

---

## Round 0.3 · accepted · Accept

I have now received two reviewers’ comment and both reviewers were satisfied with your reply and revisions from previous comments. You and your coauthors have my congratulations. Thank you for choosing PeerJ as a venue for publishing your research work and I look forward to receiving more of your work in the future.

Tsung-Min Hung, Ph.D.
PeerJ editor
Distinguished professor
Department of Physical Education
National Taiwan Normal University

Reviewer 1 ·

Basic reporting

No comments.

Experimental design

No comments.

Validity of the findings

No comments.

Additional comments

The authors have refined the manuscript that is now appropriate for publishing.

Reviewer 2 ·

Basic reporting

1.The abstract accurately described the submission.
2.The authors clearly defined a research problem and topic on this paper. Also, this paper linked theory and practice in an important way.

Experimental design

The paper defined the measures and identify how they effectively operationalize the study variables. Also,this paper provided reliability and validity data for all measures.

Validity of the findings

Already put more discussions on the validity generalization for the psychometric properties of the perceived stress scale (PSS).

Additional comments

Literature, results & analysis are effectively synthesised and results are valid. Analysis and reporting of results creates insight into theory and practice.